# Relevance of Unsupervised Metrics in Task-Oriented Dialogue for Evaluating Natural Language Generation

## Abstract

Automated metrics such as BLEU are widely used in the machine translation literature. They have also been used recently in the dialogue community for evaluating dialogue response generation. However, previous work in dialogue response generation has shown that these metrics do not correlate strongly with human judgment in the non task-oriented dialogue setting. Task-oriented dialogue responses are expressed on narrower domains and exhibit lower diversity. It is thus reasonable to think that these automated metrics would correlate well with human judgment in the task-oriented setting where the generation task consists of translating dialogue acts into a sentence. We conduct an empirical study to confirm whether this is the case. Our findings indicate that these automated metrics have stronger correlation with human judgments in the task-oriented setting compared to what has been observed in the non task-oriented setting. We also observe that these metrics correlate even better for datasets which provide multiple ground truth reference sentences. In addition, we show that some of the currently available corpora for task-oriented language generation can be solved with simple models and advocate for more challenging datasets.

## 1 Introduction

Rule-based and template-based dialogue response generation systems have been around for a long time (Axelrod, 2000; Elhadad, 1992). Even today, many task-oriented dialogue systems deployed in production are rule-based and template-based. These systems do not scale with increasing domain complexity and maintaining the increasing number of templates becomes cumbersome. In the past, Oh & Rudnicky (2000) proposed a corpus-based approach for Natural Language Generation (NLG) for task-oriented dialogue systems. Other statistical approaches were proposed using tree-based models and reinforcement learning (Walker et al., 2007; Rieser & Lemon, 2009). Recently, deep-learning based approaches (Wen et al., 2015b; Sharma et al., 2017; Lowe et al., 2015; Serban et al., 2016) have shown promising results for dialogue response generation.

The automated evaluation of machine-generated language is challenging and an important problem for the natural language processing community. The most widely used automated metrics currently are word-overlap based metrics such as BLEU (Papineni et al., 2002), METEOR (Banerjee & Lavie, 2005) which were proposed originally for machine translation. While these metrics were shown to correlate well with manual human evaluation in machine translation tasks, previous studies showed that this is not the case in non-task oriented dialogue (Liu et al., 2016). This is explained by the fact that for the same context (*e.g.* a user utterance), responses in dialogue have more diversity. Word-overlap metrics are unable to capture semantics and thus, can lead to poor scores even for appropriate responses. Human evaluation in this case is the most reliable metric. However, human judgments are expensive to obtain and not readily available at all times.

Task-oriented dialogue systems are employed in narrower domains (*e.g.* booking a restaurant) and responses do not have as much diversity as in the non-task oriented setting. Another important difference is that in the non-task oriented setting, response generation is often performed *end-to-end*, which means that the model takes as input the last user utterance and potentially the dialogue history and it outputs the next system answer. In the task-oriented setting, on the other hand, the

language generation task is often seen as a translation step from an abstract representation of a sentence to the sentence itself. As a consequence, automated metrics which compare a generated sentence to a reference sentence might be more appropriate and correlate with human judgments. In this paper, we:

- study the correlation between human judgments and several unsupervised automated metrics on two popular task-oriented dialogue datasets,
- introduce variants of existing models and evaluate their performance on these metrics

We find that the automated metrics have stronger correlation with human judgments in the task-oriented setting than what has been observed in the non task-oriented setting. We also observe that these metrics correlate even more in the presence of multiple reference sentences.

## 2 RELATED WORK

Liu et al. (2016) did an empirical study to evaluate the correlation between human scores and several automated word-overlap metrics as well as embedding-based metrics for dialogue response generation. They observed that these metrics, though widely used in the literature, had only weak correlation with human judgments in the non task-oriented dialogue NLG setting.

In terms of supervised NLG evaluation metrics, Lowe et al. (2017) proposed the ADEM model which trains a hierarchical recurrent neural network in a supervised manner to predict human-like scores. This learned score was shown to correlate better with human judgments than any other automated metric. However, the drawback of this approach is the requirement for expensive human ratings.

Li et al. (2016) proposed to use reinforcement learning to train an *end-to-end* dialogue system. They simulate a dialogue between two agents and use a policy gradient algorithm with a reward function which evaluates specific properties of the responses generated by the dialogue system.

In the adversarial setting, Kannan & Vinyals (2016) train a recurrent neural network discriminator to differentiate human-generated responses from model-generated responses. However, an extensive analysis of the viability and the ease of standardization of this approach is yet to be conducted. Li et al. (2017), apart from adversarially training dialogue response models, propose an independent adversarial evaluation metric *AdverSuc* and a measure of the model's reliability called *evaluator reliability error*. Drawbacks of these approaches are that they are model-dependent. Adversarial methods might be promising for task-oriented dialogue systems but more research needs to be conducted on their account.

Most of the work described so far has been done in the non task-oriented dialogue setting as there has been prior work indicating that automated metrics do not correlate well with humans in that setting. There has not yet been any empirical validation that these conclusions also apply to the task oriented setting. Research in the task oriented setting has mostly made use of automated metrics such as BLEU and human evaluation (Wen et al., 2015b; Sharma et al., 2017; Dušek & Jurcicek, 2016).

## 3 METRICS

This section describes the set of automatic metrics whose correlation with human evaluation is studied. We consider first word-overlap metrics and then embedding-based metrics. In all that follows, when multiple references are provided, we compute the similarity between the prediction and all the references one-by-one, and then select the maximum value. We then average the scores across the entire corpus.

### 3.1 WORD-OVERLAP BASED METRICS

### 3.1.1 BLEU

The BLEU metric (Papineni et al., 2002) compares $n$-grams between the candidate utterance and the reference utterance. The BLEU score is computed at the corpus-level and relies on the following

modified precision:

$$p_n = \frac{\sum\limits_{C \in \{Candidates\}} \sum\limits_{n-gram \in C} Ct_{clip}(n-gram)}{\sum\limits_{C' \in \{Candidates'\}} \sum\limits_{n-gram' \in C'} Ct_{clip}(n-gram')}$$

where {Candidates} are the candidate answers generated by the model and $Ct_{clip}$ is the clipped count for the $n$-gram which is the number of times the $n$-gram is common to the candidate answer and the reference answer clipped by the maximum number of occurrences of the $n$-gram in the reference answer. The BLEU-N score is defined as:

$$\text{BLEU-N} = \text{BP} \exp(\sum\limits_{n}^{N} \omega_n \log(p_n))$$

where $N$ is the maximum length of the $n$-grams (in this paper, we compute BLEU-1 to BLEU-4), $\omega$ is a weighting that is often uniform and BP is a brevity penalty. In this paper we report the BLEU score at the corpus level but we also compute this score at the sentence level to analyze its correlation with human evaluation.

### 3.1.2 METEOR

The METEOR metric (Banerjee & Lavie, 2005) was proposed as a metric which correlates better at the sentence level with human evaluation. To compute the METEOR score, first, an alignment between the candidate and the reference sentences is created by mapping each unigram in the candidate sentence to 0 or 1 unigram in the reference sentence. The alignment is not only based on exact matches but also stem, synonym, and paraphrase matches. Based on this alignment, unigram precision and recall are computed and the METEOR score is:

$$\text{METEOR} = F_{mean}(1-p)$$

where $F_{mean}$ is the harmonic mean between precision and recall with the weight for recall 9 times a high as the weight for precision, and $p$ is a penalty.

### 3.1.3 ROUGE

ROUGE (Lin, 2004) is a set of metrics that was first introduced for summarization. We compute ROUGE-L which is an F-measure based on the Longest Common Subsequence (LCS) between the candidate and reference utterances.

## 3.2 EMBEDDING BASED METRICS

We consider another set of metrics which compute the cosine similarity between the embeddings of the predicted and the reference sentence instead of relying on word overlaps.

### 3.2.1 SKIP-THOUGHT

The Skip-Thought model (Kiros et al., 2015) is trained in an unsupervised fashion and uses a recurrent network to encode a given sentence into an embedding and then decode it to predict the preceding and following sentences. The model was trained on the BookCorpus dataset (Zhu et al., 2015). The embeddings produced by the encoder have a robust performance on semantic relatedness tasks. We use the pre-trained Skip-Thought encoder provided by the authors[1].

We also compute other embedding-based methods which have been used as evaluation metrics for measuring human correlation in recent literature (Liu et al., 2016) for non task-oriented dialogue in Sections 3.2.2, 3.2.3, and 3.2.4.

### 3.2.2 EMBEDDING AVERAGE

This metric computes a sentence-level embedding by averaging the embeddings of the words composing this sentence:

$$\bar{e}_C = \frac{\sum_{w \in C} e_w}{|\sum_{w' \in C} e_{w'}|}.$$

---

[1]https://github.com/ryankiros/skip-thoughts

In this equation, the vectors $e_w$ are embeddings for the words $w$ in the candidate sentence $C$.

### 3.2.3 VECTOR EXTREMA

Vector extrema (Forgues et al., 2014) computes a sentence-level embedding by taking the most extreme value of the embeddings of the words composing the sentence for each dimension of the embedding:

$$e_{rd} = \begin{cases} \max_{w \in C} e_{wd} & \text{if } e_{wd} > |\min_{w' \in C} e_{w'd}| \\ \min_{w \in C} e_{wd} & \text{otherwise.} \end{cases}$$

In this equation, $d$ is an index over the dimensions of the embedding and $C$ is the candidate sentence.

### 3.2.4 GREEDY MATCHING

Greedy matching does not compute a sentence embedding but directly a similarity score between a candidate $C$ and a reference $r$ (Rus & Lintean, 2012). This similarity score is computed as follows:

$$G(C, r) = \frac{\sum w \in C \max_{\hat{w} \in r} cos\_sim(e_w, w_{\hat{w}})}{|C|}$$
$$GM(C, r) = \frac{G(C, r) + G(r, C)}{2}.$$

Each word in the candidate sentence is greedily matched to a word in the reference sentence based on their embeddings' cosine similarity. The score is an average of these similarities over the number of words in the candidate sentence. The same score is computed by reversing the roles of the candidate and reference sentences and the average of the two scores gives the final similarity score.

## 4 RESPONSE GENERATION MODELS

This section presents the different natural language generation models that we use in this study. All of these models take as input a set of dialogue acts (Austin, 1962) potentially with slot types and slot values and translate this input into an utterance. An example input is `inform(food = Chinese)` and a corresponding output would be "I am looking for a Chinese restaurant.". In this example, the dialogue act is `inform`, the slot type is *food*, and the slot value is *Chinese*.

### 4.1 RANDOM

Given a dialogue act with one or more slot types, the random model finds all the examples in the training set with the same dialogue act and slots (while ignoring slot values) and it randomly selects its output from this set of reference sentences. The datasets that we experiment on have some special slot values such as "yes", "no", and "don't care". Since the model ignores all slot values, these special cases are not properly handled, which results in slightly lower performance than what we could get by spending more time hand-engineering the model's behavior for these values.

### 4.2 LSTM

This model consists of a recurrent LSTM (Hochreiter & Schmidhuber, 1997) decoder. The dialogue acts and slot types are first encoded as a binary vector whose length is the number of possible combinations of dialogue acts and slot types in the dataset. We refer to this binary vector as the Dialogue Act ($DA$) vector. The $DA$ vector for a given set of dialogue acts is a binary vector over the fused dialogue act-slot types, e.g., `INFORM-FOOD`, `INFORM-COUNT`, etc.

This binary vector is given as input to the decoder at each time-step of the LSTM. The decoder then outputs a delexicalized sentence. A delexicalized sentence contains placeholders for the slot values. An example is "I am looking for a FOOD restaurant.". The values for the delexicalized slots (the type of food in this example) are then directly copied from the input.

### 4.3 DELEX-SC-LSTM

This model uses the same architecture as the LSTM model presented in the previous section except that it uses sc-LSTM (Wen et al., 2015b) units in the decoder instead of LSTM units. We call this

model the "delex-sc-LSTM"[2]. As in the previous model, the input $DA$ vector only encodes acts and delexicalized slots. It does not contain any information about the slot value.

By providing this model the same $DA$ vector input as the one given to the LSTM model, we can directly study if the additional complexity of the sc-LSTM unit's reading gate provides significant improvement over the small-sized task-oriented dialogue datasets which are currently available.

### 4.4 HIERARCHICAL-LEX-DELEX-SC-LSTM

This model is a variant of the "ld-sc-LSTM" model proposed by Sharma et al. (2017) which is based on an encoder-decoder framework. We call our model "hierarchical-lex-delex-sc-LSTM"[3].

We present the encoder in Figure 1. The encoder consists of a hierarchical LSTM with $N_e$ time-steps, where $N_e$ is the number of non-zero entries in the $DA$ vector. Each time-step of the encoder encodes one dialogue act's delexicalized and lexicalized slot-value pair (e.g. (INFORM-FOOD, 'Chinese')). The delexicalized act-slot part is encoded as a one-hot vector which we refer to as $DA_t$. $DA_t$ is constructed by masking all except the $t^{\text{th}}$ dialogue act in the $DA$ vector[4]. The lexicalized value part is en-

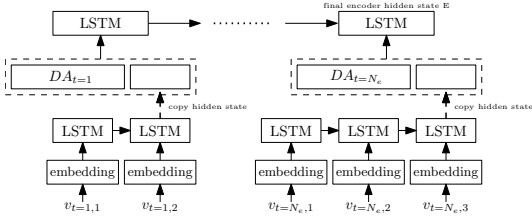

Figure 1: Encoder of the hld-scLSTM model

coded by an LSTM encoder which shares parameters across all time-steps and operates over the word-embeddings of the lexicalized values $v_{t,i}$. Our model differs from the "ld-sc-LSTM" model in that we use an LSTM encoder over the word-embeddings instead of computing the mean of the word-embeddings. The final hidden state of this LSTM is concatenated with $DA_t$ and is given as input to the upper LSTM (see Figure 1). The final hidden state of the upper LSTM is then provided to the decoder as input. This is another difference from the "ld-sc-LSTM" as that uses the mean of all the hidden states of the encoder instead, which, in our experiments, did not perform as well as using just the final hidden state $E$.

The decoder is described in Figure 2. It is the same as in the "ld-sc-LSTM" model. At each time-step, it takes as input the encoder output $E$, the $DA$ vector, and the word-embedding of the word generated at the previous time-step. The $DA$ vector is also additionally provided to the sc-LSTM cell in order for it to be regulated by its reading gate as described in Wen et al. (2015b).

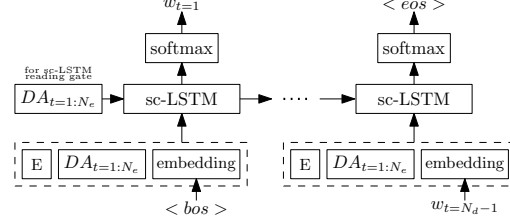

Figure 2: Decoder of the hld-scLSTM model

## 5 EXPERIMENTS

### 5.1 DECODING

During training, at each time-step, we use the ground truth word from the previous time-step. The model thus learns to generate the next word given the previous one. On the other hand, to generate sentences during test time, we use beam search. The first word input to the generator is a special token $< bos >$ which indicates the beginning of the sequence. Decoding is stopped if we reach a specified maximum number of time-steps or if the model outputs a special token $< eos >$ which indicates the end of the sequence. We also use a slot error rate penalty, similarly to Wen et al. (2015b), to re-rank the sentences generated with beam search. We use this method for all three of the LSTM, d-scLSTM, and hld-scLSTM models for fairness.

---

[2]We will also refer to it as "d-scLSTM".

[3]We will also refer to it as"hld-scLSTM".

[4]also referred to as $DA_{t=1:N_e}$

Table 1: Performance comparison across models on word-overlap based automated metrics

| | DSTC2 | | | | | | Restaurants | | | | | |
|---|---|---|---|---|---|---|---|---|---|---|---|---|
| | B-1 | B-2 | B-3 | B-4 | M | R_L | B-1 | B-2 | B-3 | B-4 | M | R_L |
| Gold | **1.00** | **1.00** | **1.00** | **1.00** | **1.00** | **1.00** | **1.00** | **1.00** | **1.00** | **1.00** | **1.00** | **1.00** |
| Random | 0.875 | 0.843 | 0.822 | 0.807 | 0.564 | 0.852 | 0.872 | 0.813 | 0.765 | 0.721 | 0.504 | 0.796 |
| LSTM | 0.900 | 0.879 | 0.863 | 0.851 | 0.610 | 0.888 | 0.982 | 0.966 | 0.949 | 0.932 | 0.652 | 0.944 |
| d-scLSTM | 0.880 | 0.850 | 0.828 | 0.812 | 0.578 | 0.874 | 0.980 | 0.964 | 0.948 | 0.931 | 0.654 | 0.945 |
| hld-scLSTM | **0.909** | **0.890** | **0.878** | **0.870** | **0.624** | **0.899** | **0.985** | **0.978** | **0.970** | **0.962** | **0.704** | **0.965** |

Table 2: Performance comparison across models on sentence-embedding based automated metrics

| | DSTC2 | | | | | Restaurants | | | |
|---|---|---|---|---|---|---|---|---|---|
| | Skip Thought | Embedding Average | Vector Extrema | Greedy Matching | | Skip Thought | Embedding Average | Vector Extrema | Greedy Matching |
| Gold | **1.00** | **1.00** | **1.00** | **1.00** | | **1.00** | **1.00** | **1.00** | **1.00** |
| Random | 0.906 | 0.981 | 0.910 | 0.947 | | 0.843 | 0.957 | 0.905 | 0.930 |
| LSTM | **0.946** | 0.985 | 0.935 | 0.962 | | 0.945 | **0.997** | 0.986 | 0.991 |
| d-scLSTM | 0.925 | 0.984 | 0.926 | 0.957 | | 0.948 | **0.997** | 0.986 | 0.991 |
| hld-scLSTM | 0.932 | **0.987** | **0.942** | **0.964** | | **0.968** | **0.997** | **0.989** | **0.993** |

Similarly to the LSTM model, the d-scLSTM and hld-scLSTM generate delexicalized sentences, *i.e.*, they generate slot tokens instead of slot values directly. These slot tokens are replaced with slot values in a post-processing step which is a fairly common step in task-oriented dialogue NLG literature.

## 5.2 EVALUATION

In NLG tasks, improvements in automated metric scores are most commonly used to demonstrate improvement in the generation task. However, these metrics have been shown to only weakly correlate with human evaluation in the non task-oriented dialogue setting (Liu et al., 2016) and hence are not considered reliable measures of improvement. Human evaluation is considered the metric of choice, but human ratings are expensive to obtain. The ease of computing these automated metrics and their availability for rapid prototyping has lead to their widespread adoption.

We evaluate the models described in the previous section on the DSTC2 (Henderson et al., 2014) and the Restaurants datasets (Wen et al., 2015a) using these automated metrics. These datasets are some of the only available resources for studying NLG for task-oriented dialogue. The DSTC2 dataset contains dialogues between human users and a dialogue system in a restaurant domain. The dataset is annotated with dialogue acts, slot type, and slot values. The NLG component of the dialogue system used for data collection is templated. The Restaurants dataset was specifically proposed for NLG and provides, for a set of dialogue acts with slot types and slot values, two sentences generated by humans.

We present the results of our experiments in Table 1 and Table 2. The scores of all the models on these automated metrics are very high. This indicates that there is significant word overlap between the generated and the reference sentences and that the NLG task on these datasets can be solved with a simple model such as the LSTM model. In effect, Table 1 shows that the LSTM model performs comparably to the d-scLSTM model based on the word-overlap metrics. This can be explained by the fact that the d-scLSTM model has more parameters and might suffer from overfitting issues on these relatively small datasets.

The hld-scLSTM is considered to consistently outperform the other models based on the word-overlap metrics. As explained by Sharma et al. (2017), this improvement results from the model's access to the lexicalized slot values, due to which it can take into account the grammatical associations of the generated words near the output tokens, thereby generating higher quality sentences. However, Table 2 shows that sentence-embedding based metrics judge all the models except the random one to perform quite similarly with again, very high performance scores.

In the next section, we add human evaluation for these models on these datasets.

## 5.3 HUMAN RATING COLLECTION

We randomly selected 20 dialogue acts from the test set of each dataset. For each of these contexts, we presented 5 sentences to the evaluators: the gold response provided in the test set and the responses generated by the four models described in Section 4. These sentences

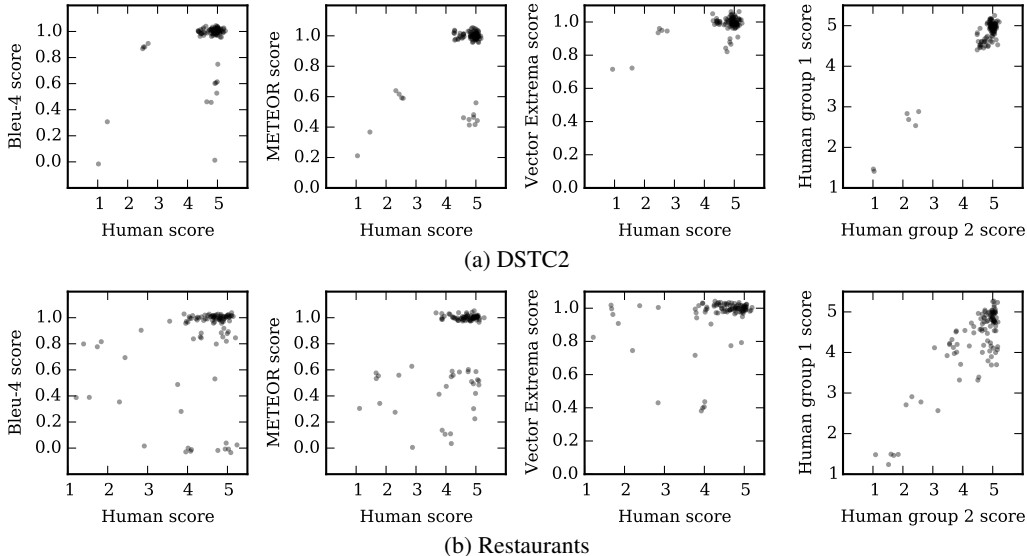

(a) DSTC2

(b) Restaurants

Figure 3: Scatter plots for correlation of some automated metrics with human evaluation for (a) the DSTC2 dataset, and (b) the Restaurants dataset. Random gaussian noise $\mathcal{N}(0, 0.1)$ has been added to data points along the human score axis and $\mathcal{N}(0, 0.02)$ has been added to the automated metric score's axis to aid visualization of overlapping data points. Transparency has been added for the same effect.

were randomly shuffled and not presented in the same order. We invited 18 human users to score each of these 100 sentences on a Likert-type scale of 1 to 5. The users were asked to rate the responses depending on how appropriate they were for the specified dialogue acts. A score of 1 was the lowest score, meaning that the response was not appropriate at all whereas a score of 5 meant that the sentence was highly appropriate.

We computed Cohen's kappa scores (Cohen, 1960) between the human users in pairs of two. We removed 7 users who had kappa scores less than 0.1 and used the remaining 11 users for the correlation study. The kappa scores are presented in Table 3. Most of the user pairs have a Cohen's $\kappa > 0.3$ which indicates fair agreement between users (Viera et al., 2005).

Table 3: Pairwise Cohen's kappa scores for the 11 human users

| $\kappa$ | # pairs | % pairs |
|---|---|---|
| >0.1 | 55/55 | 100.0 % |
| >0.2 | 40/55 | 72.7 % |
| >0.3 | 28/55 | 50.9 % |
| >0.4 | 19/55 | 34.5 % |
| >0.5 | 8/55 | 14.5 % |
| >0.6 | 0/55 | 0.0 % |

### 5.4 CORRELATION BETWEEN AUTOMATED METRICS AND HUMAN SCORES

We present the correlation between the automated metrics and our collected human ratings in Table 4. We measure human *v.s.* human correlation by randomly splitting the human users into two groups. The results indicate that in most cases, human scores correlate the best with other human scores. Except in the case of the Spearman correlation for BLEU-N scores, we can see that there is a positive correlation between the automated metrics and the human scores for these task-oriented datasets, which contrasts with the non task-oriented dialogue setting where Liu et al. (2016) observed no strong correlation trends.

A likely explanation for the negative Spearman correlation values for BLEU-N is that there is only one gold reference per context in the DSTC2 dataset. The Restaurants dataset, on the other hand, provides two gold references per context. Having multiple gold references increases the likelihood that the generated response will have significant word-overlap with one of the reference responses.

We present scatter plots for some of the metrics presented in Table 4 in Figure 3. We observe that all the metrics correlate very well with humans on high scoring examples. As it can be seen in the scatter plots, most of the sentences are given the maximal score of 5 by the human evaluators. This

Table 4: Correlation of automated metrics with human evaluations scores

| Metric | DSTC2 | | | | Restaurants | | | |
|---|---|---|---|---|---|---|---|---|
| | Spearman | p-value | Pearson | p-value | Spearman | p-value | Pearson | p-value |
| Bleu 1 | -0.317 | <0.005 | 0.583 | <0.005 | 0.069 | 0.494 | 0.277 | 0.005 |
| Bleu 2 | -0.318 | <0.005 | 0.526 | <0.005 | 0.091 | 0.366 | 0.166 | 0.099 |
| Bleu 3 | -0.318 | <0.005 | 0.500 | <0.005 | 0.109 | 0.280 | 0.223 | 0.026 |
| Bleu 4 | -0.318 | <0.005 | 0.461 | <0.005 | 0.105 | 0.296 | 0.255 | 0.010 |
| METEOR | 0.295 | <0.005 | 0.582 | <0.005 | 0.353 | <0.005 | 0.489 | <0.005 |
| ROUGE_L | 0.294 | <0.005 | 0.448 | <0.005 | 0.346 | <0.005 | 0.382 | <0.005 |
| Skip Thought | 0.528 | <0.005 | 0.086 | 0.397 | 0.284 | <0.005 | 0.364 | <0.005 |
| Embedding Average | 0.295 | <0.005 | 0.485 | <0.005 | 0.423 | <0.005 | 0.260 | 0.009 |
| Vector Extrema | 0.299 | <0.005 | 0.624 | <0.005 | 0.446 | <0.005 | 0.287 | <0.005 |
| Greedy Matching | 0.295 | <0.005 | 0.572 | <0.005 | 0.446 | <0.005 | 0.325 | <0.005 |
| Human | 0.810 | <0.005 | 0.984 | <0.005 | 0.653 | <0.005 | 0.857 | <0.005 |

confirms our previous observation that the available corpora for task-oriented dialogue NLG task are not very challenging and a simple LSTM-based model can output high-quality responses.

Overall, among the word overlap based automated metrics, METEOR consistently correlates with human evaluation on both datasets. These results confirm the original findings by Banerjee & Lavie (2005) who showed that METEOR had good correlation with human evaluation in the machine translation task. The comparison with machine translation is highly relevant in the task-oriented setting because the NLG model essentially learns to translate the abstract representation of a sentence into a sentence. It is a translation task contrary to the non task-oriented setting where the NLG model needs to decide and output a new sentence based on the last sentence typed by a user and dialogue history. Therefore, automated metrics coming from the machine translation literature are more adequate in our case than in the non-task oriented case as shown by Liu et al. (2016).

It is interesting to see that METEOR correlates well with human evaluation consistently. This can be explained by the fact that even though METEOR does not rely on word embeddings, it includes notions of synonymy and paraphrasing when computing the alignment between the candidate and reference utterances.

## 6 DISCUSSION

We evaluated several natural language generation models trained on the DSTC2 and the Restaurants datasets based on several automated metrics. We also performed human evaluation on the model-generated responses and our study shows that human evaluation is a much more reliable metric compared to the others. Among the word-overlap based automated metrics, we found that the METEOR score correlates the most with human judgments and we suggest using METEOR for task-oriented dialogue natural language generation instead of BLEU. We also observe that these metrics are more reliable in the task-oriented dialogue setting compared to the general, non task-oriented one due to the limited possible diversity in the task-oriented setting. Also, as observed by Galley et al. (2015), we can see that word-overlap based metrics correlate better with human evaluation when multiple references are provided, as in the Restaurants dataset. Otherwise, as in the case of DSTC2 which only provides one reference sentence per example, we observe that all the BLEU-N metrics negatively correlate with human evaluation on Spearman correlation.

As has been observed in the machine translation literature, using beam search improves the quality of generated sentences significantly compared to stochastic sampling. For similar models, our results show improvement in the automated metrics' scores compared to Wen et al. (2015b) who used stochastic sampling for decoding instead of beam search.

Wen et al. (2015b) did not use the slot error rate penalty with the vanilla LSTM model in their experiments. After adding the penalty in our case, we observe that the vanilla LSTM-based model performs as well as the delexicalized semantically-controlled LSTM model. This suggests that the added complexity introduced by the sc-LSTM unit does not offer a significant advantage for these two datasets.

High performance on automated metrics, achieved by our models on the DSTC2 and the Restaurants datasets lead us to conclude that these datasets are not very challenging for the NLG task. The task-oriented dialogue community should move towards using larger and more complex datasets, which have been recently announced, such as the Frames dataset (El Asri et al., 2017) or the E2E NLG Challenge dataset (Novikova et al., 2016).

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
