# OpenReview forum: "Relevance of Unsupervised Metrics in Task-Oriented Dialogue for Evaluating Natural Language Generation"
_ICLR.cc/2018/Conference — Reject_

### Official Review · AnonReviewer3 · 2017-11-20
**Useful conclusion; not enough novel contribution to warranty publication**

**Rating:** 4
**Confidence:** 4

**Review:**

This paper's main thesis is that automatic metrics like BLEU, ROUGE, or METEOR is suitable for task-oriented natural language generation (NLG). In particular, the paper presents a counterargument to "How NOT To Evaluate Your Dialogue System..." where Wei et al argue that automatic metrics are not correlated or only weakly correlated with human eval on dialogue generation. The authors here show that the performance of various NN models as measured by automatic metrics like BLEU and METEOR is correlated with human eval.

Overall, this paper presents a useful conclusion: use METEOR for evaluating task oriented NLG. However, there isn't enough novel contribution in this paper to warrant a publication. Many of the details unnecessary: 1) various LSTM model descriptions are unhelpful given the base LSTM model does just as well on the presented tasks 2) Many embedding based eval methods are proposed but no conclusions are drawn from any of these techniques.

---

> ### Author Response · Authors · 2018-01-05
> **response from authors**
>
> Thank you for the valuable feedback!
>
> 1) Our results on metrics contradict previous work by Wen et al. (2015) which observed much lower BLEU scores for the same datasets. We presented all the base models so that we could have a clearer comparison between their results and ours.
> 2) We couldn't find any clear correlation trends for the embedding based metrics. We will report that in the paper. In future work, we will look at larger datasets as mentioned above to AnonReviewer2 and AnonReviewer1 where there might possibly be clearer trends.

---

### Official Review · AnonReviewer1 · 2017-11-27
**borderline**

**Rating:** 5
**Confidence:** 4

**Review:**

1) This paper conducts an empirical study of different unsupervised metrics' correlations in task-oriented dialogue generation. This paper can be considered as an extension of Liu, et al, 2016 while the later one did an empirical study in non-task-oriented dialogue generation.

2)My questions are as follows:
i) The author should give the more detailed definition of what is non-task-oriented and task-oriented dialogue system. The third paragraph in the introduction should include one use case about non-task-oriented dialogue system, such as chatbots.
ii) I do not think DSTC2 is good dataset here in the experiments. Maybe the dataset is too simple with limited options or the training/testing are very similar to each other, even the random could achieve very good performance in table 1 and 2. For example, the random solution is only 0.005 (out of 1) worse then d-scLSTM, and it also has a close performance compared with other metrics. Even the random could achieve 0.8 (out of 1) in BLEU, this is a very high performance.
iii) About the scatter plot Figure 3, the authors should include more points with a bad metric score (similar to Figure 1 in Liu 2016).
iv) About the correlations in figure b, especially for BLEU and METEOR, I do not think they have good correlations with human's judgments.
v) BLEU usually correlates with human better when 4 or more references are provided. I suggest the authors include some dataset with 4 or more references instead of just 2 references.

---

> ### Author Response · Authors · 2018-01-05
> **response from authors**
>
> Thank you for the valuable feedback!
>
> 2)
> (i) We will make these changes.
> (ii) We agree with this analysis. We also found the DSTC2 dataset to be very simple for the NLG task. In future work we will be using the datasets from El Asri et al. (2017) and Novikova et al. (2016) as mentioned by AnonReviewer2 as well.
> (iii) We also include all the points but due to these task-oriented dialog datasets datasets being very simple most of these are overlapping in the upper right cluster.
> (iv) Most of the points being in the upper right cluster does distort the correlation values.
> (v) The dataset by Novikova et al. (2016) has larger number of references. We will use that dataset in future work.

---

### Official Review · AnonReviewer2 · 2017-11-28
**Useful but incremental contribution to NLG/dialog metrics research**

**Rating:** 5
**Confidence:** 3

**Review:**

The authors present a solid overview of unsupervised metrics for NLG, and perform a correlation analysis between these metrics and human evaluation scores on two task-oriented dialog generation datasets using three LSTM-based models. They find weak but statistically significant correlations for a subset of the evaluated metrics, an improvement over the situation that has been observed in open-domain dialog generation.
Other than the necessarily condensed model section (describing a model explained at greater length in a different work) the paper is quite clear and well-written throughout, and the authors' explication of metrics like BLEU and greedy matching is straightforward and readable. But the novel work in the paper is limited to the human evaluations collected and the correlation studies run, and the authors' efforts to analyze and extend these results fall short of what I'd like to see in a conference paper.
Some other points:
1. Where does the paper's framework for response generation (i.e., dialog act vectors and delexicalized/lexicalized slot-value pairs) fit into the landscape of task-oriented dialog agent research? Is it the dominant or state-of-the-art approach?
2. The sentence "This model is a variant of the “ld-sc-LSTM” model proposed by Sharma et al. (2017) which is based on an encoder-decoder framework" is ambiguous; what is apparently meant is that Sharma et al. (2017) introduced the hld-scLSTM, not simply the ld-scLSTM.
3. What happens to the correlation coefficients when exact reference matches (a significant component of the highly-rated upper right clusters) are removed?
4. The paper's conclusion naturally suggests the question of whether these results extend to more difficult dialog generation datasets. Can the authors explain why the datasets used here were chosen over e.g. El Asri et al. (2017) and Novikova et al. (2016)?

---

> ### Author Response · Authors · 2018-01-05
> **response from authors**
>
> Thank you for the valuable feedback!
>
> 1. Most of the research in task oriented dialog generation research uses dialog / speech acts and slots as they significantly help. It is expensive to collect these labels and there has been recent work on doing task-oriented dialog generation end-to-end without these labels. However we focus on NLG in a modular framework instead of end-to-end dialog generation and acts and slots are the dominant methodology in that area.
> 2. We'll update this in the paper. They did not introduce the hld-lscLSTM model.
> 3. We haven't looked at that because most of the points are actually within that cluster but this could be an interesting analysis to improve the system.
> 4. This work was finished earlier than the conference submission period and at that time these datasets were very recent. We will be using these datasets in future work as they are much bigger and diverse than earlier datasets.

---

### Decision · Program_Chairs · 2018-01-29
**ICLR 2018 Conference Acceptance Decision**

**Decision:**

Reject

**Comment:**

This paper tackles a very important problem: evaluating natural language generation. The paper presents an overview of existing unsupervised metrics, and looks at how they correlate with human evaluation scores. This is important work and the empirical conclusions are useful to the community, but the datasets used are too limited and the authors agree it would be better to use newer bigger and more diverse datasets suggested by reviewers for drawing more general conclusions. This work would indeed be much stronger if it relied on better, more recent datasets; therefore publication as is seems premature.